# SL-VC: A Benchmark and Automated Framework for Separation Logic Verification Condition Proving

Hanyang Wang [* 1]    Xiwei Wu [* 1]    Qinxiang Cao [1]

## Abstract

Formal verification of system software with complex heap manipulations remains challenging. Standard automated solvers frequently fail to discharge separation logic verification conditions even when correct specifications like loop invariants are provided, forcing verification engineers to manually construct proofs. While large language models (LLMs) have shown promise in proof synthesis, specialized approaches for separation logic remain unexplored. To bridge this gap, we introduce SL-VC (Separation Logic Verification Conditions), a benchmark of 289 verification conditions from textbook implementations of data structures and algorithms together with real-world C code, including the LiteOS kernel's linked list library and the mini-gmp library. Our evaluation reveals that general-purpose LLMs and existing LLM-based Rocq provers struggle to effectively discharge these verification conditions. To address this challenge, we propose SPLIT (Split spatial and pure Proving with LLM-frIendly Tactics), a novel framework that enables predictable proof state transitions through an LLM-friendly tactic library, combined with a two-stage workflow that separates spatial and pure reasoning to align with separation logic semantics. Experimental results on SL-VC demonstrate that SPLIT consistently outperforms existing approaches, showing that LLM-assisted proof synthesis is a promising solution for separation logic verification of real-world system software.

*Equal contribution  [1]Shanghai Jiao Tong University, Shanghai, China.  Correspondence to: Qinxiang Cao <caoqinxiang@sjtu.edu.cn>.

*Proceedings of the 43rd International Conference on Machine Learning*, Seoul, South Korea. PMLR 306, 2026. Copyright 2026 by the author(s).

## 1. Introduction

Program verification is vital for ensuring software reliability, especially as AI-driven code generation becomes ubiquitous. Current research in AI for program verification primarily focuses on synthesizing loop invariants and specifications. To check whether a provided invariant is valid, the verification tool generates logical formulas called verification conditions (VCs), which must be proved to establish correctness. The prevailing workflow assumes that once a correct invariant is generated, the underlying verification engine (typically an SMT solver) can automatically check its validity. However, we argue that this assumption collapses when applied to system-level software involving complex heap manipulation described by separation logic (Reynolds, 2002). Unlike numerical domains, separation logic describes complex heap manipulations where standard SMT solvers frequently fail to discharge VCs, even when the provided invariants are indeed valid. To help with this, in traditional verification workflows (without AI assistance), verification engineers are forced to either (1) craft extensive intermediate assertions to decompose the proof steps for automated solvers (for example, verifying ContikiOS's linked list library of 176 lines of C code takes about 1400 lines of annotations (Blanchard et al., 2018)), or (2) export the VCs to interactive theorem provers (ITPs) like Rocq to construct the proof manually. This reliance on heavy manual intervention exposes a critical reality: discharging VCs is not a trivial automated check, but a non-trivial task that requires rigorous reasoning.

While SMT solvers struggle with these complex heap-reasoning tasks, there are strong indicators that modern AI approaches can bridge this gap. First, the existence of rule-based automation tools, such as RefinedC (Sammler et al., 2021), demonstrates that separation logic reasoning, while complex, follows traceable patterns and structural rules. Although these tools often require heavy manual annotation, their underlying mechanics prove that the deduction is systematic. Second, large-scale verification efforts, such as the formal verification of the CertiKOS kernel (Gu et al., 2016) and seL4 kernel (Klein et al., 2009), show that these VCs can be resolved under expert guidance. These evidences suggest that the reasoning strategies used by experts contain a latent distribution that large language models (LLMs) can

learn. By capturing these underlying patterns, AI has the potential to automate these labor-intensive proofs, scaling beyond the limits of rigid rules or manual effort.

To realize this potential, the research community requires high-quality data and specialized infrastructure. In this paper, we present a holistic framework to advance AI reasoning in this domain. Our contributions are:

1. We construct **SL-VC** (Separation Logic Verification Conditions), a benchmark suite of 289 VCs from textbook implementations of data structures and algorithms together with real-world C code, including the doubly linked list library from the LiteOS kernel (Gu et al., 2021)[1] and the mini-gmp library[2], targeting complex heap manipulations that generate computationally intractable VCs for SMT solvers, which establishes a new testbed for evaluating the deductive capabilities of AI models.

2. We design an **LLM-friendly tactic library** for Rocq that makes proof steps predictable for LLM reasoning. Unlike existing human-friendly tactics that perform aggressive automation, our library decomposes reasoning into explicit, granular operations, facilitating LLM's internal chain-of-thought. This library enables researchers evaluating on our benchmark to choose between LLM-friendly or human-friendly tactics.

3. We propose **SPLIT** (Split spatial and pure Proving with LLM-frIendly Tactics), a two-stage workflow that structurally separates spatial reasoning from pure conditions solving, aligning with the semantic structure of separation logic. Combined with our tactic library, this framework significantly improves the automated VC discharging performance on our benchmark.

**Remark.** It is worth clarifying why we ground our framework in ITPs rather than relying solely on automated SMT solvers. While SMT solvers are powerful, their "black-box" nature renders their behavior unpredictable, often forcing even human experts to rely on trial-and-error experimentation to test their capabilities. This opacity provides little feedback upon failure, making it difficult for models to learn from mistakes. In contrast, ITPs require explicit, step-by-step proofs, offering superior interpretability and a richer signal for learning. Furthermore, the expressive power of ITPs allows our framework to verify a broader class of programs that exceed the limits of SMT solvers.

---

[1]LiteOS is Huawei's lightweight IoT operating system designed for resource-constrained embedded devices.

[2]mini-gmp is a minimal implementation of a subset of the GNU Multiple Precision Arithmetic Library (GMP), available at https://gmplib.org/.

## 2. Background

We illustrate the verification workflow through a concrete example: verifying the `length` function that computes the length of a singly linked list. The function takes a pointer to the head of a list and returns an integer representing the number of nodes. Listing 1 shows the implementation annotated with formal specifications.

*Listing 1.* The `length` function with QCP specifications.

```
1  int length(struct list *p)
2  /*@ With l
3      Require Zlength(l)<=INT_MAX ∧ sll(p,l)
4      Ensure ret=Zlength(l) ∧ sll(p@pre,l) */
5  { int n = 0;
6    /*@ Inv ∃ l1 l2,
7        l = l1++l2 ∧ n = Zlength(l1) ∧
8        sllseg(p@pre,p,l1) * sll(p,l2) */
9    while (p) { n++; p = p→next; }
10   return n; }
```

### 2.1. Annotating the Program with Specifications

We use the Qualified C Programming Verifier (QCP) (Wu et al., 2025), which allows programmers to embed formal specifications directly in C source code using special comments with the marker `/*@ ... */`.

An *assertion* is a logical formula that describes what must be true about the program state at a particular point. In separation logic, assertions consist of two components: the *pure part*, which expresses heap-independent logical propositions such as arithmetic constraints and equalities, and the *spatial part*, which describes heap structures such as memory ownership and pointer relationships. In QCP, assertions follow the syntax `P ∧ ... ∧ Q * ... * R`, where propositions joined by ∧ form the pure part and those joined by * form the spatial part. Here * denotes the *separating conjunction*: the assertion $P * Q$ holds when the heap can be split into two disjoint regions, one satisfying $P$ and the other satisfying $Q$. The assertion `emp` denotes the empty heap assertion in separation logic, meaning that no memory is allocated.

Lines 2–4 specify the function specification. `With l` introduces a logical variable `l` representing the abstract content of the list. A precondition (after `Require`) describes what must hold before the function executes: the pure part `Zlength(l)<=INT_MAX` constrains the list length, while the spatial part `sll(p,l)` states that pointer `p` points to a well-formed singly linked list with mathematical content `l`. `sll(p,l)` is formally and recursively defined as:

```
sll(p,nil) := p=NULL ∧ emp
sll(p,a::l):= ∃ y, p.data=a * p.next=y
                    * sll(y,l)
```

A postcondition (after `Ensure`) describes what is guaranteed when the function returns: the return value equals the

list length and the original list remains intact, where `p@pre` denotes the value of `p` at function entry.

Lines 7–9 specify the loop invariant, an assertion that must hold before and after each iteration. The spatial part `sllseg(p@pre,p,l1) * sll(p,l2)` states that the heap consists of two disjoint regions: a list segment from the initial pointer to the current pointer containing `l1`, and the remaining list containing `l2`. `sllseg(p,q,l)` is formally and recursively defined as:

```
sllseg(p,q,nil) := p=q ∧ emp
sllseg(p,q,a::l):= ∃ y, p.data=a * p.next=y
                     * sllseg(y,q,l)
```

The pure part specify that `l = l1++l2` (the original list is reconstructed by concatenating the two partitions) and `n = Zlength(l1)` (the counter tracks the traversed length).

## 2.2. Generating Verification Conditions

Given the annotated program, the QCP verification tool performs forward symbolic execution to generate VCs—logical formulas that must be proved to establish program correctness. Each VC takes the form $P \vdash Q$, meaning "if the current state satisfies $P$, then $Q$ must be derivable." If all generated VCs hold, the program is correct with respect to its specification.

For the `length` function, QCP generates several VCs at critical program points. A key one is `length_entail_wit_1` (Equation 1), which verifies that when entering the loop, the initial state satisfies the loop invariant.

$$\forall p,l. \quad \texttt{Zlength}(l) \leq \texttt{INT\_MAX} \land \texttt{sll}(p,l)$$
$$\vdash \quad \exists l1,l2.\, l = l1{+}{+}l2 \land 0 = \texttt{Zlength}(l1) \quad (1)$$
$$\land \texttt{sllseg}(p,p,l1) * \texttt{sll}(p,l2)$$

Intuitively, this VC asks: given the condition that the heap contains a list `sll(p,l)` with bounded length, can we find witness values `l1` and `l2` such that the loop invariant holds before the first iteration? The answer is `l1 = nil` (no nodes traversed yet) and `l2 = l` (the entire list remains to be traversed).

## 2.3. Proving Verification Conditions

VCs are exported to the Rocq theorem prover, where users interactively construct proof scripts using QCP's original human-friendly tactic library. Proving such VCs in separation logic requires two kinds of reasoning:

**Spatial Reasoning.** The goal of spatial reasoning is to transform heap predicates (e.g., `sll` and `sllseg`) so that the spatial assertions on both sides of $\vdash$ can be matched and cancelled. The ultimate objective is to reduce the goal to `emp ⊢ emp`, indicating that all heap resources have been properly accounted for.

**Pure Reasoning.** The goal of pure reasoning is to discharge pure logical goals. For example, in Equation 1, this involves finding witnesses `l1` and `l2` to establish `l = l1++l2` and `Zlength(l1) = 0`. Such goals involve only mathematical reasoning without heap manipulation, making them similar to the proof tasks studied in neural theorem proving for mathematical formalization.

In Rocq's proof state display, the separator `------` divides the context (hypotheses, shown above) from the goal (what needs to be proved, shown below). Below the dashed line, the notation `(X/Y)` indicates that the current goal is the X-th subgoal out of a total of Y goals. Taking `length_entail_wit_1` as an example, the proof proceeds as follows:

1. `pre_process`: Introduces quantified variables and extracts pure propositions:

```
p:Z, l:list Z, H:Zlength(l)<=INT_MAX
------------------------------------
(1/1)
sll(p,l) ⊢ ∃ l1 l2, l=l1++l2 ∧
0=Zlength(l1) ∧ Zlength(l)<=INT_MAX ∧
sllseg(p,p,l1) * sll(p,l2)
```

2. `Exists nil l`: Instantiates $l_1{=}nil$ and $l_2{=}l$:

```
p:Z, l:list Z, H:Zlength(l)<=INT_MAX
------------------------------------
(1/1)
sll(p,l) ⊢ l=nil++l ∧ 0=Zlength(nil) ∧
Zlength l <= INT_MAX ∧
sllseg(p,p,nil) * sll(p,l)
```

3. `entailer!`: Cancels matching predicates and solves simple pure goals. Here it cancels `sll(p,l)`:

```
p:Z, l:list Z, H:Zlength(l)<=INT_MAX
------------------------------------
(1/1)
emp ⊢ sllseg(p,p,nil)
```

4. `simpl; entailer!`: Unfolds the definition of `sllseg`. When the list is empty, we get `sllseg(p,p,nil) = emp`, completing the proof.

The complete proof using human-friendly tactics is:

```
Proof.
  pre_process.
  Exists nil l.
  entailer!.
  simpl; entailer!.
Qed.
```

This workflow is inherently **interactive**: users execute a tactic and observe the resulting proof state to determine the next action. When the tactic successfully completes the proof, the process terminates. When the tactic leaves residue proof

goals, a human expert would carefully read and analyze the new goals, and then determine the next move. It also frequently happens that automation tactics like `entailer!` do too much simplification, turning a provable proof goal into an unprovable residue proof goal. In that case, a human expert will choose to undo the tactic and manually handle the logic that may be overly simplified. While `entailer!` can automatically discharge certain reasoning steps, tasks such as existential witness instantiation, predicate transformation, and complex logical reasoning still demand human insight.

## 3. Benchmark

To evaluate AI models on proving separation logic verification conditions, we construct SL-VC (Separation Logic Verification Conditions), a benchmark of 289 VCs from textbook implementations of data structures and algorithms together with real-world C code. Unlike existing benchmarks that focus on numerical programs, SL-VC specifically targets complex heap manipulations. Table 1 presents the overall statistics by program source.

*Table 1.* VC distribution by program source.

| Type | VCs |
| --- | --- |
| Data Structures & Algorithms | 185 |
| Real-World System Code | 104 |
| Total | 289 |

SL-VC is derived from two categories of C programs, all annotated and verified using the QCP verification framework:

**Data Structures and Algorithms.** We collect textbook-style implementations including array operations, singly linked lists (traversal, reversal, concatenation, merge sort, insertion sort), mergeable singly linked lists with a tail pointer supporting O(1) concatenation, string manipulation programs, doubly linked lists, and binary search trees (insertion, deletion, and father-pointer variants). These programs cover core data structures in computer science curricula, providing a solid foundation for evaluating models' reasoning capabilities on basic heap operations.

**Real-World System Code.** We incorporate two industrial-grade codebases: (1) LiteOS Kernel, the doubly linked list library from Huawei's IoT operating system, featuring circular linked lists with sentinel nodes; and (2) mini-gmp, a compact implementation of the GNU Multiple Precision Arithmetic Library. VCs from these real-world codebases often involve more complex invariants and corner cases, representing challenges in practical software verification.

QCP produces three files for each program: `proof_goal.v` contains all generated VCs, `proof_goal_auto.v` contains VCs discharged by QCP's SMT solver, and `proof_goal_manual.v`

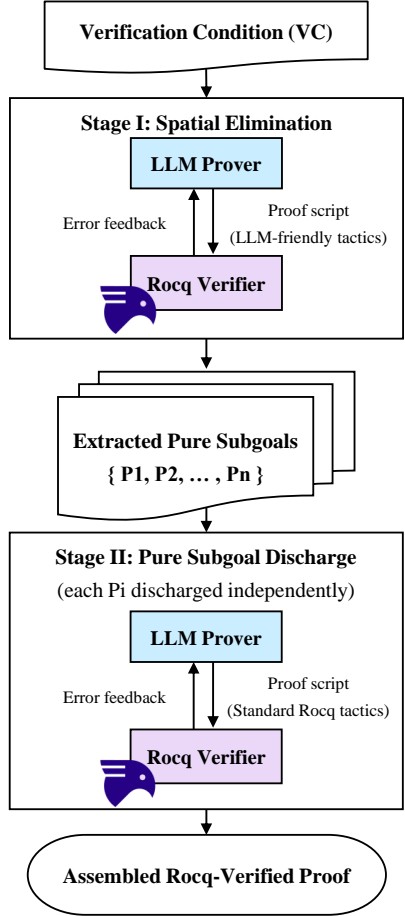

*Figure 1.* Overview of the SPLIT two-stage workflow.

contains VCs requiring manual proof. The auto/manual split is determined entirely by the tool. From the manual VCs, we remove cases solvable by the fixed preprocessing tactic `pre_process` alone.

All functions are verified by verification experts, who supply the required annotations and discharge the resulting verification conditions with the aid of human-friendly tactics.

## 4. Method

To address the challenge of verifying programs with complex memory manipulations in our benchmark, we propose a novel framework named SPLIT (Split spatial and pure Proving with LLM-frIendly Tactics). In this section, we present our approach, which coordinates large language models with a specialized LLM-friendly tactic library to conduct rigorous Separation Logic reasoning. Figure 1 illustrates the overall workflow.

### 4.1. From Human-Friendly to LLM-Friendly Tactics

We adopt a **whole-proof generation** with iterative repair workflow. The model generates the complete proof script in

one pass, submits it to the Rocq compiler, and utilizes error feedback to revise the script in subsequent rounds.

However, this non-interactive generation poses a critical challenge regarding the predictability of proof state transitions. As mentioned in Section 2, the QCP backend relies on human-friendly tactics like `entailer!` to discharge VCs. These tactics are highly efficient for human experts who can interactively observe the proof state: the expert executes a tactic, observes the resulting proof state, and determines the next action. However, in the context of whole-proof generation, these tactics are unpredictable for LLMs. Since `entailer!` performs massive, opaque simplifications—simultaneously eliminating the spatial part and solving arithmetic—the LLM cannot accurately predict the resulting proof state without actual execution. Consequently, the model's internal chain-of-thought becomes decoupled from the actual proof state, leading to hallucinations where the model generates tactics for a non-existent goal. Therefore, we propose a set of LLM-friendly tactics designed around predictable proof-state changes. The library is derived from two sources: (1) analysis of expert proof traces, where humans typically first perform spatial transformations via `sep_apply`, then use `entailer!` to eliminate most spatial predicates, and finally discharge remaining goals; and (2) functional analysis of existing tactics, decomposing multi-effect tactics like `entailer!` and refining tactics like `sep_apply` that introduce pure side effects.

To illustrate our solution, we demonstrate how our LLM-friendly tactic library handles the verification condition `length_entail_wit_1` from Section 2. After applying `pre_process`, the proof state is:

```
p_pre:Z, l:list Z, H:Zlength(l)<=INT_MAX
------------------------------------
(1/1)
sll(p_pre,l) ⊢ EX l1 l2 : list Z,
l = l1 ++ l2 ∧ 0 = Zlength(l1) ∧
Zlength(l) <= INT_MAX ∧
sllseg(p_pre,p_pre,l1) * sll(p_pre,l2)
```

Using our LLM-friendly tactic library, the proof proceeds as follows:

1. `Exists nil l`: Instantiate existential quantifiers.

   ```
   p_pre:Z, l:list Z, H:Zlength(l)<=INT_MAX
   ------------------------------------
   (1/1)
   sll(p_pre,l)
   ⊢ l = nil ++ l ∧ 0 = Zlength(nil) ∧
   Zlength(l)<= INT_MAX ∧
   sllseg(p_pre,p_pre,nil) * sll(p_pre,l)
   ```

2. `split_pure_spatial`: The right-hand side mixes pure propositions and spatial predicates. We use

`split_pure_spatial` to separate them, creating two subgoals and using bullet − to focus on each:

```
p_pre:Z, l:list Z, H:Zlength(l)<=INT_MAX
------------------------------------
(1/2)
sll(p_pre,l) ⊢
sllseg(p_pre,p_pre,nil) * sll(p_pre,l)
(2/2)
sll(p_pre,l) ⊢ l = nil ++ l ∧
0 = Zlength(nil) ∧ Zlength(l) <= INT_MAX
```

3. `cancel (sll p_pre l)`: As both sides contain `sll p_pre l`, we explicitly cancel it by `cancel (sll p_pre l)`.

```
p_pre:Z, l:list Z, H:Zlength(l)<=INT_MAX
------------------------------------
(1/1)
emp ⊢ sllseg(p_pre,p_pre,nil)
```

4. `sep_apply_r (nil_sllseg_l p_pre p_pre)`: Instead of using `simpl` which is uncontrollable for proof state changes, we apply specific lemma to unfold list segment.

```
p_pre:Z, l:list Z, H:Zlength(l)<=INT_MAX
------------------------------------
(1/1)
emp ⊢ p_pre = p_pre ∧ emp
```

5. `split_pure_spatial`: The goal has mixed spatial and pure parts. We use `split_pure_spatial` to separate them into two subgoals (bullet +). For the spatial part (emp ⊢ emp), we use `cancel emp`. For the pure part (emp ⊢ p_pre = p_pre), we use `pure_solve`. The tactic `pure_solve` is an automation tactic that attempts to discharge pure goals. It applies simplification strategies and tries to solve the goal automatically. If it succeeds, the pure goal is discharged. If it fails, the proof state remains intact and the unsolved goal is exported as a separate lemma. In this case, Rocq allows the proof to proceed but marks the pure part as "dangerously solved." Once the user proves the exported lemma, Rocq reports "safely solved." This mechanism enables our two-stage workflow (Section 4.2): spatial reasoning proceeds independently in the first stage, while the exported pure goals are collected and discharged in the second stage.

6. `pure_solve`: We return to the second subgoal using bullet −. The goal contains only pure propositions l = nil ++ l ∧ 0 = Zlength nil ∧ Zlength l <= INT_MAX which can be directly solved by `pure_solve`.

The complete proof using LLM-friendly tactics is:

```
Proof.
  pre_process.
  Exists nil l.
  split_pure_spatial.
  - cancel (sll p_pre l).
    sep_apply_r (nil_sllseg_l p_pre p_pre).
    split_pure_spatial.
    + cancel emp.
    + pure_solve.
  - pure_solve.
Qed.
```

Unlike the proof in Section 2 where `entailer!` performs multiple operations simultaneously, our LLM-friendly tactics make each logical step explicit and predictable. This granularity enables the LLM to maintain accurate internal reasoning about the proof state without executing the proof, preventing the chain-of-thought from becoming decoupled from the proof script. These two tactic sets are not mutually exclusive: one can still use the original tactics for aggressive automation to quickly close goals, while the LLM-friendly tactics expose intermediate proof states when predictability is needed. The complete list of LLM-friendly tactics is provided in Appendix A.

### 4.2. Two-Stage Semantics-Aligned Workflow

Building upon the LLM-friendly tactic library, we propose a two-stage reasoning pipeline that structurally enforces the separation between the spatial part and the pure part of assertions. As described in Section 2, assertions in separation logic naturally decompose into these two parts: the spatial part describes the heap layout, while the pure part describes logical propositions independent of the heap. This design not only divides the complex reasoning task into manageable sub-problems but also allows for optimized strategies in different domains. Algorithm 1 provides an overview of the workflow.

**Stage I: Spatial Elimination.** The first stage focuses exclusively on the spatial part of the entailment. We prompt the model to temporarily ignore arithmetic details and focus on structural matching to align the heap layout. The model utilizes a set of LLM-friendly tactics. The primary objective is to reduce the spatial entailment to $emp \vdash emp$. Once the goal is reduced to pure subgoals, the tactic `pure_solve` is invoked. This stage employs an iterative loop between an LLM prover and the Rocq verifier: the LLM generates a proof, Rocq verifies it, error feedback guides repairs, and this continues until success or budget exhaustion.

**Stage II: Pure Subgoal Discharge.** In this stage, the workflow deals with the residual verification conditions exported by `pure_solve`, which consist solely of pure subgoals. Since the remaining problems are primarily arithmetic issues, this modular design offers flexibility in solver selection. In our implementation, these pure subgoals are

---

**Algorithm 1** SPLIT Framework

**Input:** verification condition $VC$, repair budget $B$
**Output:** verified proof or failure status
  // Stage I: Spatial Elimination
  $proof \leftarrow$ LLM-Prover($VC$) with LLM-friendly tactics
  **while** not verified and $B > 0$ **do**
    $VC \leftarrow$ UpdateWithFeedback($VC$);
    $proof \leftarrow$ LLM-Prover($VC$); $B \leftarrow B - 1$;
  **end while**
  **if** not verified **then**
    **return** Failure
  **end if**
  Extract pure tasks $\{P_i\}$ from $proof$
  // Stage II: Pure Subgoal Discharge
  **for** each $P_i$ **do**
    $p_i \leftarrow$ LLM-Prover($P_i$) with standard Rocq tactics
    **while** not verified and $B > 0$ **do**
      $P_i \leftarrow$ UpdateWithFeedback($P_i$);
      $p_i \leftarrow$ LLM-Prover($P_i$); $B \leftarrow B - 1$;
    **end while**
    **if** not verified **then**
      **return** Failure
    **end if**
  **end for**
  **return** Assemble($proof, \{p_i\}$)

---

discharged by an iterative repair loop between the LLM prover and the Rocq verifier. We also evaluate CoqHammer as an alternative Stage-II solver in Appendix C.

To implement this workflow, we employ a structured in-context learning strategy. The prompt context includes: (1) the C source code generating the VC to provide semantic intent; (2) the available library of lemmas; and (3) the target VC to be discharged. The prompt template is provided in Appendix B.

## 5. Experiments

In this section, we evaluate the effectiveness of SPLIT on our proposed benchmark. Our experiments are designed to answer the following research questions:

- **RQ1 (Main Results):** How does SPLIT perform on SL-VC?

- **RQ2 (Comparison with Existing Rocq Provers):** How does SPLIT compare to existing LLM-based Rocq provers?

- **RQ3 (Component Contributions):** What are the individual contributions of the LLM-friendly tactic library and the two-stage workflow?

- **RQ4 (Generalization Across LLMs):** How does

SPLIT perform with different LLM backends?

## 5.1. Experimental Setup

**Dataset and Metric.** We conduct experiments on SL-VC (Section 3), using `length_entail_wit_1` as a few-shot prompt example. The remaining 288 VCs form the full evaluation set. For controlled comparisons that require additional prover adaptation or repeated model evaluations, we use a 207-VC core subset. We adopt **Pass@1 with repair budget** $k$ as our evaluation metric: a VC is considered solved if the generated proof is verified by Rocq within $k$ repair iterations. We report results at $k = 0$ (no repair) and $k = 10$ (up to 10 repair attempts).

**Implementation.** Unless otherwise specified, we use DeepSeek-V3.2 (temperature 0) as the LLM backend. All experiments except Rango (Thompson et al., 2025) are conducted with Rocq 8.20.1 on an Intel Core Ultra 5 225H processor with 32GB RAM. Rango uses its publicly available artifact on an NVIDIA RTX 2080 Ti GPU.

**Baselines.** Across the experiments, we compare SPLIT against three baselines. **PALM** (Lu et al., 2024) uses an LLM to generate an initial proof, then repairs it via backtracking with CoqHammer (Czajka & Kaliszyk, 2018). To adapt PALM to our domain, we directly provide the relevant lemma library as context (avoiding retrieval of verification framework infrastructure lemmas), and apply `entailer!` before invoking CoqHammer to simplify spatial predicates (10-second timeout per call). This adaptation is a best-effort attempt at fairness given that PALM was not designed for separation-logic VCs; without `entailer!`, CoqHammer faces full mixed spatial-pure goals that are substantially harder. **Rango** retrieves similar proofs and lemmas at each step to guide tactic generation. Since Rango uses a fine-tuned model trained on general Rocq projects, we include 8 example proofs covering QCP human-friendly tactics in the proof library to enable learning of domain-specific tactic usage patterns. **Human-Friendly Tactics** uses the original human-friendly tactics for one-shot proof generation, sharing the same prompt structure and repair mechanism as SPLIT. This baseline allows us to evaluate the effectiveness of our complete framework design.

## 5.2. RQ1: Main Results

Table 2 presents the main results on the full 288-VC evaluation set. SPLIT achieves a 66.0% pass rate at $k = 10$, improving over the Human-Friendly Tactics baseline's 43.4% by 22.6 percentage points.

Even without repair ($k = 0$), SPLIT achieves 14.2%, exceeding the Human-Friendly Tactics baseline's 4.5%. This demonstrates that our framework design enables the generation of higher-quality initial proofs, reducing reliance on iterative repair. At $k = 10$, the gap remains substantial

*Table 2.* Main results on the full 288-VC evaluation set.

| METHOD | PASS RATE |
|---|---|
| HUMAN-FRIENDLY TACTICS ($k = 0$) | 4.5% |
| HUMAN-FRIENDLY TACTICS ($k = 10$) | 43.4% |
| SPLIT ($k = 0$) | 14.2% |
| SPLIT ($k = 10$) | 66.0% |

*Table 3.* Comparison with existing Rocq provers on the 207-VC core subset.

| METHOD | PASS RATE |
|---|---|
| PALM | 14.0% |
| RANGO | 7.7% |
| SPLIT ($k = 0$) | 17.4% |
| SPLIT ($k = 10$) | 73.4% |

on the full evaluation set. A source-family breakdown of SPLIT's pass rate on the full evaluation set is provided in Appendix E.

## 5.3. RQ2: Comparison with Existing Rocq Provers

Table 3 compares SPLIT with existing LLM-based Rocq provers on the 207-VC core subset. SPLIT achieves a 73.4% pass rate at $k = 10$, substantially outperforming PALM and Rango.

Despite employing CoqHammer, a powerful automated reasoning tool, PALM achieves only 14.0% pass rate. This limitation stems from its backtracking repair mechanism: PALM can only backtrack and invoke hammer for repair, while in one-shot generation scenarios, human-friendly tactics struggle to sufficiently simplify complex spatial reasoning goals into small-scale subgoals amenable to hammer processing. PALM's successful cases typically involve proofs where the core difficulty lies in existential quantifier instantiation—the LLM correctly provides witnesses, after which `entailer!` and hammer can automatically complete the remaining spatial predicate elimination and pure conditions solving. However, for complex heap reasoning in SL-VC that requires multi-step spatial predicate transformations and lemma applications, this strategy proves ineffective.

Rango achieves a 7.7% pass rate. As a retrieval-augmented approach, Rango predicts the next tactic at each proof step based on retrieved similar proofs. However, Rango's fine-tuned model was not specifically trained for VC proving, separation logic reasoning, or QCP tactics. On SL-VC, Rango primarily relies on imitating similar proofs in the proof library. Due to the high diversity in spatial structures of separation logic VCs, this retrieval-based strategy struggles to generalize effectively.

Even without repair ($k = 0$), SPLIT achieves 17.4%, exceeding both existing Rocq prover baselines on the core subset. This demonstrates that our framework design enables the generation of higher-quality initial proofs, reducing reliance on iterative repair.

## 5.4. RQ3: Component Contributions

To validate the effectiveness of our LLM-friendly tactic library and two-stage workflow, we conduct ablation studies on the 207-VC core subset. Besides the Human-Friendly Tactics baseline, we establish one intermediate system: **LLM-Friendly Tactics** uses our proposed LLM-friendly tactic library but adopts a single-stage reasoning strategy. All configurations share the same prompt structure and repair mechanism. Figure 2 presents the ablation results.

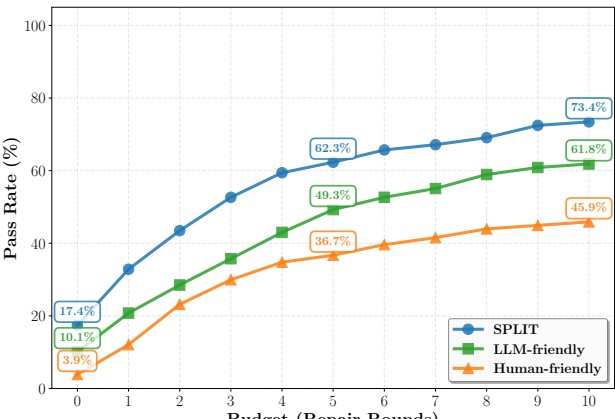

*Figure 2.* Ablation study on the 207-VC core subset, showing pass rates across different repair budgets.

As shown in the figure, introducing LLM-friendly tactics improves the pass rate at $k = 10$ from 45.9% to 61.8%, an increase of 15.9 percentage points. Building upon this, the two-stage workflow introduced in SPLIT further improves the pass rate from 61.8% to 73.4%, an increase of 11.6 percentage points. The improvements are consistent across all repair budget levels. Notably, even without repair ($k = 0$), LLM-Friendly Tactics achieves 10.1% (2.6 times the Human-Friendly Tactics baseline's 3.9%), and SPLIT further improves this to 17.4%. This demonstrates that predictable proof state transitions and structured separation of spatial and pure reasoning both contribute to higher-quality initial proofs. The gap between methods widens as more repair iterations become available, suggesting that these tactics and structured workflows facilitate more effective error correction.

Furthermore, the two-stage workflow significantly expands the solvable problem space. SPLIT uniquely solves 26 VCs, while LLM-Friendly Tactics uniquely solves only 5 and Human-friendly Tactics only 1. These SPLIT-exclusive successful VCs cover diverse complex scenarios, including BST operations (8 VCs), GMP library functions (9 VCs), queue and linked list operations (4 VCs), among others. This demonstrates that explicit separation of spatial and pure reasoning enables the model to handle complex reasoning tasks featuring both heap and arithmetic that other approaches cannot solve.

*Table 4.* Performance of SPLIT with different LLMs on the 207-VC core subset.

| LLM BACKEND | $k = 0$ | $k = 10$ |
|---|---|---|
| DEEPSEEK-V3.2 | 17.4% | 73.4% |
| QWEN3-235B-A22B | 7.7% | 51.2% |
| KIMI-K2-THINKING | 7.7% | 51.7% |

The remaining failures are concentrated in a few sources. On the 207-VC core subset, 55 VCs remain unsolved by SPLIT at $k = 10$. The top failure sources are gmp (21/52, 40.4%), sum (9/17, 52.9%), and bst_fp_delete (9/18, 50.0%). The higher failure rate in gmp is driven by VCs involving correctness of arithmetic computations, where the spatial stage often requires filling complicated existential variables and the pure stage needs complex arithmetic reasoning. By contrast, LiteOS VCs involve more structural data manipulation, where spatial transformations follow regular patterns and pure reasoning only involves list reasoning.

## 5.5. RQ4: Generalization Across LLMs

To evaluate the performance of SPLIT with different LLM backends, we test three LLMs on the 207-VC core subset. The results in Table 4 demonstrate that SPLIT maintains its effectiveness across different LLM backends, though performance varies with model capability. DeepSeek-V3.2 achieves 73.4% pass rate at $k = 10$, while Qwen and Kimi-K2-Thinking reach around 51%. A stage-level analysis shows that DeepSeek's advantage mainly comes from the spatial stage: DeepSeek solves 92.8% of spatial-stage goals, compared with 67.6% for Qwen and 64.7% for Kimi. On pure-stage goals, the gap is much smaller: DeepSeek solves 66.9%, Qwen solves 58.0%, and Kimi solves 67.9%.

# 6. Related Work

## 6.1. LLM-based Program Verification in ITPs

SynVer (Mukherjee & Delaware, 2024) synthesizes and verifies C programs in the VST framework (Cao et al., 2018) which supports separation logic. Their verification relies on SEPAUTO, an expert-crafted automation tactic; when SEPAUTO leaves unresolved goals, an LLM suggests tactics, after which SEPAUTO is invoked again iteratively. Our work differs in two aspects: (1) SynVer performs full Hoare logic derivation, where VC discharge is one component; we focus specifically on VC proving. (2) SynVer combines native VST tactics with expert-designed automation, whereas we design a tactic library specifically tailored for LLM reasoning without heavily relying on rule-based automation.

AutoRocq (Tu et al., 2026) adopts an agentic approach where an LLM autonomously interacts with Rocq, introducing a benchmark of VCs from SV-COMP (Beyer & Strejcek, 2025) programs via Frama-C (Kirchner et al., 2015). Their VCs focus on integer overflow and functional cor-

rectness properties involving integer arithmetic and logical constraints. In contrast, our benchmark targets separation logic VCs that require reasoning about heap structures and spatial predicate transformations.

NTP4VC (Xu et al., 2026) builds a broader multi-language VC benchmark from Why3/Frama-C, covering diverse VC categories. The two works are complementary: NTP4VC targets general VC proving, while SL-VC focuses on separation-logic VCs that involve memory manipulation, requiring both spatial and pure reasoning, a setting that is more realistic for C program verification but not covered by NTP4VC.

Other recent work explores LLM-based verification in ITPs from different perspectives. VeriBench (Miranda et al., 2025) and CLEVER (Thakur et al., 2025) target end-to-end verified code generation in Lean 4, where models generate implementations, specifications, and proofs from informal inputs such as Python code or natural language descriptions. In Isabelle, FVEL (Lin et al., 2024) generates lemma specifications and proofs for C code verification, while Selene (Zhang et al., 2024) focuses on proof generation for existing specifications from the seL4 microkernel verification project. None of these address separation logic or verification condition proving, which is the focus of our work.

### 6.2. Proof Automation in Rocq

Proof automation in Rocq includes traditional tools like CoqHammer (Czajka & Kaliszyk, 2018), which combines premise selection with external automated theorem provers, and LLM-based methods such as PALM (Lu et al., 2024), Rango (Thompson et al., 2025), Cobblestone (Kasibatla et al., 2025), Adapt (Lu et al., 2025), RocqStar (Khramov et al., 2025), etc.

We evaluated representative methods from both categories on our benchmark, and the results show limited effectiveness. These approaches target general-purpose theorem proving and are evaluated on benchmarks like Coq-Gym (Yang & Deng, 2019) and CoqStoq (Thompson et al., 2025), which consist of mathematical theorems and mostly lemma library in program verification. None are specifically adapted for separation logic VC proving.

### 6.3. LLM-based Annotation Generation

Another line of work uses LLMs to generate program annotations such as loop invariants, pre/postconditions and auxiliary lemmas, which are then automatically discharged by verifiers. Lemur (Wu et al., 2024) employs LLMs to propose loop invariants verified by automated solvers. AutoVerus (Yang et al., 2024), RAG-Verus (Zhong et al., 2025), and DafnyBench (Loughridge et al., 2025) target annotation

generation for Verus and Dafny, which rely on Z3 for proof discharge. For separation logic, LIG-MM (Liu et al., 2024) generates loop invariants for heap-manipulating programs, checked by an entailment solver, and Rego et al. (Rego et al., 2024) evaluate annotation generation for VeriFast. These approaches focus on annotation generation with automatic checking, whereas our work requires explicit tactic-based proofs in an ITP.

## 7. Discussion

**Scope beyond QCP.** SPLIT relies on proof goals admitting spatial/pure decomposition, not on QCP specifically. For Frama-C/WP, which is not centered on separation logic, we expect less gain from Stage I. For VST-Floyd, VST performs full Hoare-logic derivation in Rocq with separation-logic VCs arising as part of that process; QCP's frontend automates much of the Hoare-logic reasoning and only exports residual VCs. SPLIT could target the VC subproblem within VST but would require adaptation to its tactic infrastructure. QCP and VST are very similar in their framework design, and our method can apply to VST and Iris (Jung et al., 2018). In particular, the `entailer!` problem we identify is equally present in VST, since VST also relies on `entailer!` for goal simplification.

## 8. Conclusion

In this paper, we focus on the critical task of discharging verification conditions (VCs) in the program verification pipeline, particularly for programs involving separation logic assertions. To address the lack of high-quality data for this task, we first constructed SL-VC, a benchmark suite of VCs from textbook implementations of data structures and algorithms together with real-world C code. Unlike previous datasets, SL-VC specifically targets VCs regarding memory manipulations. Second, to automate the proving of VCs, we proposed SPLIT. By employing an LLM-friendly tactic library and a two-stage workflow, our framework mitigates the unpredictability of standard human-friendly tactics. Experimental results on SL-VC demonstrate that our approach significantly improves the success rate of VC discharging. We believe SL-VC will serve as a valuable resource, and our framework SPLIT offers a reliable solution for this essential step in automated program verification.

## Software and Data

The SL-VC benchmark, code, and results are archived in the SL-VC artifact (Wang et al., 2026).

## Acknowledgements

This work was supported by the National Natural Science Foundation of China under Grant No. 62472274.

## Impact Statement

This paper contributes to automating formal verification of system software, an area where manual effort has traditionally been a major bottleneck. Our benchmark and framework specifically target heap-manipulating programs common in operating systems and security-critical libraries. We anticipate positive impact through reduced verification costs and improved software quality. Like most program analysis tools, our framework could theoretically be misused, but its primary value lies in improving software safety.

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

# A. LLM-Friendly Tactic Library

This appendix provides a complete reference for the LLM-friendly tactic library used in SPLIT. Each tactic is designed to perform a single, predictable transformation of the proof state.

### 1. `pre_process`

**Purpose:** Unfolds definitions, introduces quantified variables, and extracts pure propositions from the precondition. This tactic is typically pre-applied in the proof template.

### 2. `Intros x / Intros x y ...`

**Purpose:** Introduces existentially quantified variables from the left-hand side.

**Proof State Transition:**

Before:

```
----------------------------------
(1/1)
∃ n, n > 0 ∧ sll p l ⊢ ...
```

After `Intros n`:

```
n : nat
----------------------------------
(1/1)
n > 0 ∧ sll p l ⊢ ...
```

### 3. `Intros_p H`

**Purpose:** Introduces a pure proposition `P` from the left-hand side and names it `H`.

**Proof State Transition:**

Before:

```
----------------------------------
(1/1)
n > 0 ∧ sll p l ⊢ ...
```

After `Intros_p H`:

```
H: n > 0
----------------------------------
(1/1)
sll p l ⊢ ...
```

### 4. `Exists x / Exists x y ...`

**Purpose:** Instantiates existential quantifiers on the right-hand side with concrete witness values.

**Proof State Transition:**

Before:

```
l1' : list
l2' : list
----------------------------------
(1/1)
sll p l ⊢ ∃ l1 l2, l = l1 ++ l2 ∧ ...
```

After `Exists l1' l2'`:

```
l1' : list
l2' : list
-----------------------------------
(1/1)
sll p l ⊢ l = l1' ++ l2' ∧ ...
```

## 5. `split_pure_spatial`

**Purpose:** When the right hand side has the form P ∧ ... ∧ Q * ... * R, splits the goal into two subgoals: (1) spatial part, and (2) pure part. This tactic would fail if the right hand side contains non-instantiated variables.

**Proof State Transition:**

Before:

```
-----------------------------------
(1/1)
sll p l ⊢ l = nil ++ l ∧ 0 = Zlength nil ∧ sllseg p p nil * sll p l
```

After `split_pure_spatial`:

Subgoal 1 (spatial):

```
-----------------------------------
(1/2)
sll p l ⊢ sllseg p p nil * sll p l
```

Subgoal 2 (pure):

```
-----------------------------------
(2/2)
sll p l ⊢ l = nil ++ l ∧ 0 = Zlength nil
```

## 6. `cancel P`

**Purpose:** Cancels matching spatial predicates P from both sides. Only use when the right hand side contains only spatial predicates. The predicate P must match exactly on both sides.

**Proof State Transition:**

Before:

```
-----------------------------------
(1/1)
sll p l ⊢ sllseg p p nil * sll p l
```

After `cancel (sll p l)`:

```
-----------------------------------
(1/1)
emp ⊢ sllseg p p nil
```

## 7. `pure_solve`

**Purpose:** Solves the goal if the right hand side contains only pure propositions. Otherwise, it fails.

**Proof State Transition:**

Before:

```
------------------------------------
(1/1)
sll p l ⊢ l = nil ++ l ∧ 0 = Zlength nil
```

After `pure_solve`:

```
No more subgoals.
```

## 8. `sep_apply_l`

**Purpose:** Applies a separation logic lemma to transform the left-hand side. Given a lemma `H: forall x: T, C →` `P * Q ⊢ R`, using `sep_apply_l (H x)` replaces the matching part in the left hand side with the lemma's right hand side. Requires explicit instantiation of all variables but not the premise which is considered as a pure task exported to the second stage.

**Proof State Transition:**

Given lemma:

```
Lemma sll_zero: forall x l,
  x = NULL → sll x l ⊢ l = nil ∧ emp.
```

Before:

```
p: Z
l: list
------------------------------------
(1/1)
sll p l * Q ⊢ R
```

After `sep_apply_l (sll_zero p l)`:

```
p: Z
l: list
------------------------------------
(1/1)
l = nil ∧ emp * Q ⊢ R
```

## 9. `sep_apply_r`

**Purpose:** Applies a separation logic lemma to transform the right-hand side. Given a lemma `H: forall x: T, C` `→ P * Q ⊢ R`, using `sep_apply_r (H x)` replaces the matching part in the right hand side with the lemma's left hand side. Requires explicit instantiation of all variables but not the premise which is considered as a pure task exported to the second stage.

**Proof State Transition:**

Given lemma:

```
Lemma sll_zero: forall x l,
  x = NULL → l = nil ∧ emp ⊢ sll x l.
```

Before:

```
p: Z
l: list
------------------------------------
(1/1)
Q ⊢ R * sll x l
```

After `sep_apply_r (sll_zero p l)`:

```
p: Z
l: list
-----------------------------------
(1/1)
Q ⊢ l = nil ∧ emp * R
```

## 10. `Left`

**Purpose:** When the goal has the form `P ⊢ Q || R` where `||` means disjunction, changes the goal to `P ⊢ Q`.

**Proof State Transition:**

Before:

```
-----------------------------------
(1/1)
P ⊢ Q || R
```

After `Left`:

```
-----------------------------------
(1/1)
P ⊢ Q
```

## 11. `Right`

**Purpose:** When the goal has the form `P ⊢ Q || R`, changes the goal to `P ⊢ R`.

**Proof State Transition:**

Before:

```
-----------------------------------
(1/1)
P ⊢ Q || R
```

After `Right`:

```
-----------------------------------
(1/1)
P ⊢ R
```

## 12. `Split`

**Purpose:** Splits a goal of the form `P || Q ⊢ R` into two subgoals: `P ⊢ R` and `Q ⊢ R`.

**Proof State Transition:**

Before:

```
-----------------------------------
(1/1)
P || Q ⊢ R
```

After `Split`:

Subgoal 1:

```
-----------------------------------
(1/2)
P ⊢ R
```

Subgoal 2:

```
------------------------------------
(2/2)
Q ⊢ R
```

## B. Prompt Structure

This appendix presents the structured prompt template used in SPLIT. The prompt is designed to provide comprehensive context for LLM-based proof generation, including the source program, domain-specific knowledge, and the target verification condition.

### B.1. Overall Structure

Our prompt consists of three main components, presented in the following order:

1. **Annotated C Program**: The source code that generates the verification condition

2. **Domain Library**: Natural language explanations of spatial predicates and available lemmas

3. **Target Verification Condition**: The goal to be proved with its initial proof state

### B.2. Component 1: Annotated C Program

This section provides the C source code with formal specifications to help the model understand the semantic intent behind the verification condition.

```
#### Annotated C Program

This section provides the C program from which the verification condition (VC) to be
proved originates. This will offer important insights for thinking about how to prove
this verification condition.

```C
{FUNCTION_SIGNATURE}
/*@ With {LOGICAL_VARIABLES}
    Require {PRECONDITION}
    Ensure {POSTCONDITION}
*/
{
    {FUNCTION_BODY}
    /*@ Inv {LOOP_INVARIANT} */
    while ({LOOP_CONDITION}) {
        {LOOP_BODY}
    }
    {FUNCTION_BODY}
    return {RETURN_VALUE};
}
```
```

### B.3. Component 2: Domain Library

This section provides natural language explanations of spatial predicates and lists all available lemmas with their formal statements.

```
#### {LIBRARY_NAME} Library

##### Predicate Explanations

**Spatial Predicates**:
```

```
1. `{PREDICATE_1} {PARAMS_1}`: {NATURAL_LANGUAGE_EXPLANATION_1}
2. `{PREDICATE_2} {PARAMS_2}`: {NATURAL_LANGUAGE_EXPLANATION_2}
...

##### Lemma Library

This section provides the lemmas you can use. You may use these lemmas directly
without proof. However, you must not create your own lemmas or use lemmas beyond
those listed here.

##### `{LEMMA_NAME}`
Lemma:
'''coq
Lemma {LEMMA_NAME}: forall {PARAMETERS}, {PREMISE} → {LHS} ⊢ {RHS}.
'''
...
```

### B.4. Component 3: Target Verification Condition

This section presents the verification condition to be proved, including its formal definition and the initial proof state after executing pre_process.

```
#### Goal to Prove

This section contains the goal you need to prove. In the `#### Complete Proof Code`
section of your output, you should first copy the entire content of the '''coq ... '''
code block below, and then complete the proof.

'''coq
Definition {VC_NAME} := forall {QUANTIFIED_VARIABLES}, {LHS} ⊢ {RHS}.

Lemma proof_of_{VC_NAME} : {VC_NAME}.
Proof.
  pre_process.
  (* Continue the proof here *)
'''

After executing `pre_process.`, the proof state is:

'''proof state
{CONTEXT_VARIABLES}
{HYPOTHESES}
=============================
{GOAL}
'''

Please continue your reasoning from this proof state in the `#### Proof Strategy`
section.
```

## C. CoqHammer as an Alternative Stage-II Solver

We compare the LLM-based Stage-II solver with CoqHammer on the 121 goals that entered the pure stage. The LLM solves 81/121 goals (66.9%), while CoqHammer solves 54/121 goals (44.6%). Pairwise, 51 goals are solved by both, 30 only by the LLM, and 3 only by CoqHammer.

The 30 LLM-only cases are concentrated on goals involving inductive data structures such as lists and binary trees, where CoqHammer struggles to discover the right combination of domain-specific library lemmas (e.g., app_assoc, sublist_split) and case analysis steps. The LLM currently handles these cases better. While the framework does support modular solver replacement, the improvement from CoqHammer is limited in this domain.

## D. Benchmark Scope

SL-VC contains programs that manipulate data structures and either do not contain much computation, or contain some computation (mini-gmp examples) but do not involve very difficult reasoning about the computation.

Programs with non-trivial algorithms often require very complicated loop invariants or recursion invariants. Recent research suggests that it is better to use relational Hoare logic for such programs: describing the algorithm with formal pseudocode in a theorem prover, verifying its correctness, and proving refinement between the source code and pseudocode. Proving refinement is very different from current program verification tasks.

For RBT and AVL trees, important concepts such as paths from the root to a leaf node cannot be easily formalized in the inductive data types used by existing BST verification tasks. SL-VC avoids VCs from a possibly incorrect verification approach.

Typical examples of more difficult pointer-manipulating programs include LLVM's implementation of control-flow graphs and assignment commands, where uses of variables have tagged links to their definitions and other links represent control flow. Such systems are too complicated to extract into small self-contained verification tasks.

More broadly, VCs from real programs consist of two components: memory layout properties and data properties. When the latter involves very complex reasoning, it is unclear whether such proofs should be expressed through invariants and VCs or handled through alternative methods, a frontier question in program verification.

## E. Source-Family Breakdown on the Full Evaluation Set

Table 5 reports SPLIT's pass rate at $k = 10$ by program source on the full 288-VC evaluation set.

*Table 5.* Source-family breakdown of SPLIT on the full 288-VC evaluation set ($k = 10$).

| SOURCE | PASSED / TOTAL | PASS RATE |
|---|---|---|
| SLL | 64 / 68 | 94.1% |
| MERGEABLE SLL | 17 / 22 | 77.3% |
| STRLIB | 10 / 15 | 66.7% |
| DLL | 7 / 8 | 87.5% |
| BST | 36 / 54 | 66.7% |
| ARRAY | 8 / 17 | 47.1% |
| LITEOS | 6 / 8 | 75.0% |
| MINI-GMP | 42 / 96 | 43.8% |
| TOTAL | 190 / 288 | 66.0% |

