# OpenReview forum: "SL-VC: A Benchmark and Automated Framework for Separation Logic Verification Condition Proving"
_ICML.cc/2026/Conference — ICML 2026 regular_

### Official Review · Reviewer_1dNb · 2026-03-04

**Soundness:** 3
**Presentation:** 2
**Significance:** 3
**Originality:** 3
**Overall Recommendation:** 5
**Confidence:** 4

**Summary:**

This paper addresses the problem of automatically discharging separation logic verification conditions (VCs) in interactive theorem provers (ITPs). This work makes two main contributions: (1) SL-VC, a benchmark of 208 VCs derived from C programs annotated with separation logic specifications using the QCP framework, covering data structure, algorithms, and real-world system code; and (2) SPLIT, a two-stage proof strategy that combines an LLM-friendly tactic library with a workflow that structurally separates spatial reasoning from pure subgoal reasoning. Experiments on SL-VC demonstrate that SPLIT outperforms existing LLM-based Rocq provers, achiving 73.4% pass rate with 10 repair iterations.

**Compliance With Llm Reviewing Policy:**

Affirmed.

**Final Justification:**

This paper studies an underexplored and meaningful problem: proving separation-logic verification conditions (VCs) in interactive theorem provers. I found the benchmark contribution and the proposed SPLIT strategy valuable, and the ablations provide solid support for the main design choices. The work is technically sound and relevant to both neural theorem proving and program verification.

My main concerns were about benchmark construction, missing fine-grained result breakdowns, the scope of generalizability beyond QCP, and some overstated framing. The rebuttal addressed these concerns well. The authors clarified the benchmark selection process, provided the requested breakdowns by VC source/type, discussed the applicability of SPLIT beyond QCP more carefully, and agreed to revise the framing around “real-world C programs.” These clarifications increased my confidence in the paper.

I am therefore raising my recommendation to Accept, with the expectation that the key clarifications and additional analyses from the rebuttal will be incorporated into the revised manuscript.

**Key Questions For Authors:**

See Weaknesses

**Limitations:**

The authors should consider discussing the restriction to QCP as the VC source and its implications for generalizability.

**Strengths And Weaknesses:**

## Strengths

- S1. **The paper targets a genuine and underexplored gap.** General-purpose neural theorem proving benchmarks (e.g., miniF2F) focus on mathematical reasoning, while separation logic VC proving requires qualitatively different reasoning such as frame rule application, spatial predicate manipulation. Few works have specifically targeted this combination.
- S2. **The ablation study convincingly supports each design decision.** The paper cleanly isolates the contribution of the LLM-friendly tactic library and the two-stage workflow, and the improvements are consistent across all repair budget levels.
- S3. **The background and motivating example are well-presented.** Section 2 provides a concrete walkthrough of the SL verification workflow, clearly situating the specific bottleneck this paper aims to address. This is helpful for an AI audience less familiar with program verification.

## Weaknesses

- W1. **The benchmark selection criteria are not explained.** Based on the supplementary material, the VCG pipeline generates over 650 proof obligations in total (correct me if I'm wrong), of which 208 are included in the benchmark. The selection ratio varies across categories: for instance, 140 safety_wit obligations are generated but only 5 are included, while entail_wit and return_wit are retained at much higher rates (~86% and ~82%). The paper provides no explanation of the selection criteria. Were the excluded obligations deemed trivial and thus unrepresentative? Or were some excluded because they could not be manually proved? If the latter, this is a significant concern: harder VCs that resist manual proof are precisely the ones that would most meaningfully challenge LLM capabilities and resist benchmark saturation. The authors should clearly describe the selection process and justify the inclusion/exclusion decisions.

- W2. **No fine-grained breakdown of results by VC type or source.** The benchmark contains VCs of qualitatively different characters: purely spatial entailments, purely arithmetic subgoals, and mixed obligations with both spatial and pure parts. Similarly, VCs from textbook and those from real-world system code likely differ substantially in difficulty. Reporting a single aggregate pass rate of 73.4% obscures where SPLIT succeeds and where it fails. A breakdown by VC category and by source would significantly strengthen the empirical contribution.

- W3. **The generalizability of SPLIT beyond QCP is not discussed.** SPLIT seems to be presented as a general methodological contribution for SL VC proving. However, it is unclear whether and how SPLIT would apply to VCs generated by other frameworks such as Frama-C (which uses WP-style VC generation and targets SMT solvers rather than ITPs) or VST-Floyd. The authors should discuss the scope of applicability of their tactic library and two-stage workflow.

- W4. **The adaptation of PALM may not be fair.** The paper notes that `entailer!` can over-simplify proof goals, potentially turning provable goals into unprovable residues. Yet PALM is adapted by first calling `entailer!` before invoking CoqHammer. The authors should clarify whether this is the most reasonable adaptation of PALM to the SL-VC domain, or whether alternative adaptations were considered. As stated, this baseline may be handicapped in a way that inflates SPLIT's apparent advantage.

- W5. **The notation emp is not explained.** The paper uses emp throughout without definition. While standard in the SL literature, this is an AI venue where many readers may be unfamiliar with this notation. A brief definition would improve accessibility.

- W6. **The characterization of SL-VC as derived from "real-world C programs" is potentially overstated.** Table 1 shows that 148 out of 208 VCs (71%) come from "Data Structures & Algorithms", which the paper itself describes as "textbook-style implementations". Only 60 VCs (29%) are drawn from actual industrial codebases (LiteOS and mini-gmp). The abstract and introduction repeatedly emphasize that SL-VC is "derived from real-world C programs", which may give readers the impression that the benchmark is predominantly grounded in industrial code. The authors should either revise the framing to more accurately reflect the composition of the benchmark, or provide a stronger justification for why textbook-style implementations qualify as "real-world" in this context.

- Comment. **Relationship to concurrent work NTP4VC not discussed. (Not a weakness)** A recent work, [NTP4VC (Xu et al., ICLR 2026)](https://openreview.net/forum?id=MfDyickxQA), also constructs a benchmark for LLM-based VC proving in ITPs and evaluates existing models on it. While the two works differ in important ways, the authors are encouraged to discuss the relationship explicitly. This would help readers better understand the unique contribution of SL-VC and SPLIT in the broader NTP-for-verification landscape, and would also be an opportunity to discuss SPLIT's potential applicability beyond the QCP setting.

---

> ### Author Rebuttal · Authors · 2026-03-30
>
> We thank Reviewer 1dNb for the thorough feedback, especially the recognition of a genuine underexplored gap (S1) and the well-supported ablation study (S2). We address each point below.
>
> > Weakness 1. The benchmark selection criteria are not explained. The selection ratio varies across categories (e.g., 140 safety_wit generated but only 5 included).
>
> R1. QCP produces three files per program: `proof_goal.v` (all VCs), `proof_goal_auto.v` (discharged by QCP's SMT solver), and `proof_goal_manual.v` (requiring manual proof). The auto/manual split is determined entirely by the tool. From the manual VCs, we removed a small number solvable by a fixed preprocessing tactic (`pre_process`) alone, yielding the final 208 VCs.
>
> The variation in retention rates reflects automatic filtering, not cherry-picking. `safety_wit` are simple arithmetic checks easily discharged by the SMT solver (135/140 auto-resolved); `entail_wit` and `return_wit` involve spatial transformations the solver cannot handle, so they are retained at higher rates.
>
> > Weakness 2. No fine-grained breakdown of results by VC type or source.
>
> R2. Thank you for the suggestion. We computed statistics by VC source, VC type, and whether the goal contains pure subtasks.
>
> By source family:
>
> | Source | Passed / Total | Pass rate |
> | --- | --- | --- |
> | sll | 64 / 68 | 94.1% |
> | dll | 7 / 8 | 87.5% |
> | bst | 36 / 54 | 66.7% |
> | array | 8 / 17 | 47.1% |
> | LiteOS | 6 / 8 | 75.0% |
> | mini-gmp | 31 / 52 | 59.6% |
>
> By VC type:
>
> | VC type | Passed / Total | Pass rate |
> | --- | --- | --- |
> | which_implies_wit | 40 / 44 | 90.9% |
> | return_wit | 71 / 98 | 72.4% |
> | entail_wit | 39 / 59 | 66.1% |
> | safety_wit | 1 / 5 | 20.0% |
>
> By pure subtask presence: spatial-only 71/86 (82.6%), with pure subtasks 81/121 (66.9%). SPLIT performs well on sll, dll, and which_implies_wit, but worse on array, mini-gmp, bst, and mixed spatial-pure cases.
>
> > Weakness 3. Generalizability of SPLIT beyond QCP is not discussed.
>
> R3. SPLIT relies on proof goals admitting spatial/pure decomposition, not on QCP specifically. For Frama-C/WP, which is not centered on separation logic, we expect less gain from Stage I. For VST-Floyd, VST performs full Hoare-logic derivation in Rocq with separation-logic VCs arising as part of that process; QCP's frontend automates much of the Hoare-logic reasoning and only exports residual VCs. SPLIT could target the VC subproblem within VST but would require adaptation to its tactic infrastructure.
>
> > Weakness 4. The adaptation of PALM may not be fair. PALM is adapted by first calling entailer! before CoqHammer.
>
> R4. We use the following example to illustrate how a proof goal can be destroyed: in `sll(NULL, l) |-- sll(NULL, l) && [| length(l) = 0 |]`, cancelling `sll(NULL, l)` without first deriving `l = nil` via `prop_apply` makes the pure goal unprovable. Neither tactic set fully prevents this, but LLM-friendly tactics offer more correction opportunities through atomic steps. For example, the LLM-friendly workflow uses `split_pure_spatial` and `dump_pre_spatial` to isolate and focus on the pure proof step by step, while `entailer!` collapses this into one opaque step. The more atomic design gives the model more intermediate states to detect a wrong path. These cases are a minority (13/208 proofs use `prop_apply`).
>
> For PALM: such cases require whole-proof resampling, which conflicts with PALM's single-proof-with-symbolic-repair philosophy. We add `entailer!` before CoqHammer so PALM operates on simpler residual goals; without it, CoqHammer faces full mixed spatial-pure goals that are substantially harder. Our adaptation is a best-effort attempt at fairness given that PALM was not designed for separation-logic VCs.
>
> > Weakness 5. The notation `emp` is not explained.
>
> R5. Thank you for pointing this out. `emp` denotes the empty heap assertion in separation logic (no memory is allocated). We will add a definition in the revised paper.
>
> > Weakness 6. The characterization of SL-VC as derived from "real-world C programs" is overstated.
>
> R6. Thank you for pointing out this ambiguity. Our use of "real-world" was intended to contrast with programs that only involve a few variables and no heap reasoning. Programs involving memory manipulation are more realistic and closer to actual C codebases. We acknowledge that this wording can be misleading and will revise to "textbook implementations of data structures and algorithms together with real-world C code."
>
> > Comment 1. Relationship to concurrent work NTP4VC not discussed.
>
> R7. Thank you for pointing out this connection. We will discuss NTP4VC explicitly. The two works are complementary: NTP4VC builds a broader multi-language VC benchmark from Why3/Frama-C, covering diverse VC categories. Our work focuses on separation-logic VCs that involve memory manipulation, requiring both spatial and pure reasoning, a setting that is more realistic for C program verification but not covered by NTP4VC.

---

> > ### Author Rebuttal · Reviewer_1dNb · 2026-04-03
> >
> > Thanks for the detailed and helpful rebuttal. The clarifications substantially address my concerns, especially regarding benchmark construction, result breakdown, and the framing of the contribution. I am inclined to raise my score to 5, provided that the revised manuscript incorporates these important clarifications and the additional analyses discussed in the rebuttal.

---

### Official Review · Reviewer_be4U · 2026-03-12

**Soundness:** 3
**Presentation:** 3
**Significance:** 3
**Originality:** 3
**Overall Recommendation:** 4
**Confidence:** 3

**Summary:**

The paper introduces Separation Logic-VC, a benchmark of 208 verification conditions derived from real-world C programs, and SPLIT, an LLM-assisted framework for discharging these conditions in the Rocq theorem prover. The key idea is that standard human-friendly tactics are too opaque for LLM-based whole-proof generation, as they perform too many simultaneous operations making proof state transitions unpredictable. SPLIT addresses this through an LLM-friendly tactic library that decomposes reasoning into explicit granular steps, and a two-stage workflow that separates spatial heap reasoning from pure arithmetic discharge, aligning with the semantic structure of separation logic.

**Compliance With Llm Reviewing Policy:**

Affirmed.

**Final Justification:**

Authors have addressed my concerns. I encourage the authors to add the new benchmarks and discussion around them to the final version. Also, the discussion around non-interactive/interactive setup

**Key Questions For Authors:**

1. Section 4.1 “whole-proof generation with iterative repair workflow” → The LLM-friendly tactic library addresses the unpredictability of whole-proof generation, but do you think this problem may disappear entirely if the LLM interacted with Rocq step-by-step and received immediate state feedback after each tactic, as in systems like Copra [1]. Can you clarify whether the granular tactic design is a fundamental advancement for separation logic verification across other frameworks like VST or Iris, or primarily a workaround for the absence of a live interactive agent.
2. The two-stage decomposition assumes that spatial and pure reasoning can be cleanly separated. For VCs where the two are semantically interleaved what changes would be required to the current framework?

[1] An In-Context Learning Agent for Formal Theorem-Proving. Thakur et al.

**Limitations:**

Currently the paper has not addressed the limitations of the approach

**Strengths And Weaknesses:**

Strengths:
1. The problem is well-motivated and the background section does an excellent job walking through the full verification workflow with a concrete example.
2. The benchmark construction is thoughtful: pairing textbook data structures with real industrial code (LiteOS, mini-gmp) nicely demonstrates that the approach scales beyond toy programs to the kinds of heap manipulations that actually arise in systems software.
3. The ablation study is well-designed, cleanly attributing gains to the tactic library and the two-stage workflow separately, and the improvements are consistent across all repair budget levels rather than appearing only at the extremes.

Weakness:

1. The paper does not analyze the 26.6% of VCs that remain unsolved at k=10. Are these systematically harder, or do they disproportionately come from LiteOS or mini-gmp? A breakdown would significantly help.
2. The benchmark of 208 VCs is small by current standards,  LeanDojo, for instance, contains over 100k theorems. More concerning is that the proposed approach already solves 73.4% by prompting alone, which raises questions about the benchmark's long-term utility for driving progress in finetuning or RL-based approaches.
3. Section 4.2 claims that Stage 2 pure goals can be offloaded to external solvers like Z3, motivating the modular design. However, no experiments with alternative solvers are provided. An evaluation with Z3 as the Stage 2 solver would help justify the modularity of the approach

---

> ### Author Rebuttal · Authors · 2026-03-30
>
> We thank Reviewer be4U for the thoughtful evaluation, particularly the recognition of the well-designed ablation study and benchmark construction. We address each point below.
>
> > Weakness 1. The paper does not analyze the 26.6% of VCs that remain unsolved. Are these systematically harder, or do they disproportionately come from LiteOS or mini-gmp?
>
> R1. There are 55 unsolved cases. The breakdown by source:
>
> | VC source group | Failed / Total | Failure rate |
> | --- | --- | --- |
> | Data Structures & Algorithms | 32 / 147 | 21.8% |
> | Real-World (LiteOS) | 2 / 8 | 25.0% |
> | Real-World (mini-gmp) | 21 / 52 | 40.4% |
>
> The top failure sources are: gmp (21/52, 40.4%), sum (9/17, 52.9%), bst_fp_delete (9/18, 50.0%). The higher real-world failure rate is driven by mini-gmp, not LiteOS. The two programs differ in nature: mini-gmp VCs involve correctness of arithmetic computations, where the spatial stage often requires filling complicated existential variables and the pure stage needs complex arithmetic reasoning that the LLM cannot get right in one attempt. LiteOS VCs, by contrast, involve more structural data manipulation (e.g., inserting a task into a pending list), where spatial transformations follow regular patterns and pure reasoning only involves list reasoning that the LLM handles well.
>
> > Weakness 2. The benchmark of 208 VCs is small by current standards. The approach already solves 73.4%, raising questions about long-term utility.
>
> R2. We acknowledge that 208 VCs are modest by ML-benchmark standards. However, we do not think the benchmark is close to saturation: 55 cases remain unsolved, with categories such as `mini-gmp`, `bst`, `array`, and goals with nontrivial pure subtasks remaining substantially harder. Moreover, as discussed in our response to Reviewer n52k (R1), separation-logic VCs are inherently expensive to construct compared to non-memory-manipulation benchmarks, and some benchmarks in new directions start at this scale.
>
> > Weakness 3. No experiments with alternative solvers (e.g., Z3) for Stage 2.
>
> R3. Thank you for the suggestion. We ran an additional experiment replacing the LLM in Stage 2 with CoqHammer. Among the 121 goals that entered the pure stage:
>
> | Setting | Passed / Total | Pass rate |
> | --- | --- | --- |
> | Original SPLIT (LLM) | 81 / 121 | 66.9% |
> | CoqHammer | 54 / 121 | 44.6% |
>
> Pairwise: 51 solved by both, 30 only by the LLM, 3 only by CoqHammer. The 30 LLM-only cases are concentrated on goals involving inductive data structures such as lists and binary trees, where CoqHammer struggles to discover the right combination of domain-specific library lemmas (e.g., `app_assoc`, `sublist_split`) and case analysis steps. The LLM currently handles these cases better. While the framework does support modular solver replacement, the improvement from CoqHammer is limited in this domain.
>
> > Question 1. Do you think this problem may disappear entirely if the LLM interacted with Rocq step-by-step? Is the granular tactic design a fundamental advancement or primarily a workaround for the absence of a live interactive agent?
>
> R4. Whole-proof generation reduces model query overhead. In practice, current code agents commonly generate a whole proof first, then repair specific subgoals through finer interaction, a pattern also observed in the Rocq-MCP team's proof-log analysis on Rocq miniF2F [1]. We therefore believe optimizing whole-proof generation is worthwhile even though tactic-by-tactic interaction can in principle avoid maintaining proof state in CoT.
>
> > Question 2. For VCs where spatial and pure reasoning are semantically interleaved, what changes would be required?
>
> R5. Spatial and pure reasoning can be semantically intertwined. For example, in `sll(NULL, l) |-- sll(NULL, l) && [| length(l) = 0 |]`, directly cancelling `sll(NULL, l)` loses the fact needed (`l = nil`) to prove `length(l) = 0`. However, syntactically the pure and spatial parts of the postcondition are explicitly separated, so the current framework does not require structural changes. But such tricky cases suggest that global feedback from pure to spatial stages could help; this is a direction for future improvement on the framework.
>
> [1] Putnam 2025 Problems in Rocq using Opus 4.6 and Rocq-MCP.

---

> > ### Author Rebuttal · Reviewer_be4U · 2026-04-04
> >
> > Thank you for the detailed response!
> >
> > W2) I understand that constructing separation-logic benchmarks is expensive, and I appreciate that point. However, this does not yet establish the benchmark’s durability as a target for future finetuning or RL-based methods. I do not view the lack of closed-source evaluations as a flaw by itself. However, it does limit the strength of the claim that the benchmark will remain challenging as model capabilities improve.
> >
> > Q1) The rebuttal explains why whole-proof generation remains practically relevant, since current proof agents often follow a generate-then-repair workflow. I find that point reasonable. However, it does not fully answer the underlying question of whether the LLM-friendly tactic library is a fundamental contribution to separation-logic proving, or mainly a design choice tailored to their non-interactive setup. In particular, the rebuttal does not show whether these tactics would still provide a clear advantage in a live step-by-step interaction setting, nor whether the idea transfers beyond their current QCP/Rocq workflow to other frameworks such as VST or Iris

---

> > > ### Author Response · Authors · 2026-04-06
> > >
> > > Thank you for the follow-up questions! We address them below.
> > >
> > > **R-W2.** We appreciate your insightful question, which pushes us to think more deeply about our data collection. Currently, our dataset contains programs that manipulate data structures and (1) do not contain much computation, or (2) do contain some computation (mini-gmp examples), but do not involve very difficult reasoning about the computation.
> > >
> > > We did consider other sources but chose to exclude them.
> > >
> > > (1) *Programs with non-trivial algorithms.* In these cases, very complicated loop invariants or recursion invariants are needed. However, recent research suggests that it is better to use relational Hoare logic to verify such programs: (i) describing the algorithm with formal pseudocode in a theorem prover, (ii) verifying its correctness, and (iii) proving refinement between the source code and pseudocode. Proving refinement is very different from current program verification tasks (although recent work shows they can use the same tool).
> > >
> > > (2) *More Binary Search Trees.* Existing results of BST verification tasks usually formalize the tree as an inductive data type and complete the verification using separation logic predicates on these inductively defined trees. However, we found that this approach and existing works mainly focus on naive implementations of unbalanced BST. For RBT and AVL trees, important concepts like "paths from the root to a leaf node" cannot be easily formalized in such inductive data types. We chose not to include such programs in our dataset because we did not want to provide VCs from a possibly incorrect verification approach.
> > >
> > > (3) *More difficult programs with pointer manipulation.* Typical examples of this category are LLVM's implementation of control flow graphs and assignment commands. Uses of variables have tagged links to their definitions, while other links are used to represent control flow. We chose not to include such programs in our dataset simply because LLVM is too complicated and we failed to extract out a small self-contained verification task from it.
> > >
> > > More broadly, VCs from real programs consist of two components: memory layout properties and data properties. When the latter involves very complex reasoning, it is unclear whether such proofs should be expressed through invariants and VCs or handled through alternative methods, a frontier question in program verification. This informed our exclusion decisions.
> > >
> > > We plan to incorporate this discussion above into the paper revision. We also identified three additional sources: (1) mergeable sll (22 VCs), a boxed singly linked list with a tail pointer supporting O(1) concatenation; (2) strlib (15 VCs), string manipulation programs; and (3) additional mini-gmp functions on addition (44 VCs), outer-layer wrappers that do not directly manipulate arrays, but wrap inner data and handle different cases through case analysis. We consider these encapsulation typical data structure operations too and add them. We have now added 81 new VCs, increasing the benchmark from 208 to 289 VCs. The new evaluation (k = repair budget) is:
> > >
> > > | Method | k | Original (207 VCs) | New (81 VCs) | Combined (288 VCs) |
> > > | --- | ---: | ---: | ---: | ---: |
> > > | Human-Friendly | 0 | 3.9% | 6.2% | 4.5% |
> > > | Human-Friendly | 10 | 45.9% | 37.0% | 43.4% |
> > > | SPLIT | 0 | 17.4% | 6.2% | 14.2% |
> > > | SPLIT | 10 | 73.4% | 46.9% | 66.0% |
> > >
> > > The source-level breakdown at k=10: new gmp 22.7%/25.0%, mergeable sll 54.5%/77.3%, strlib 53.3%/66.7% (Human-Friendly/SPLIT). The new VCs are harder (combined SPLIT k=10: 66.0%), supporting long-term utility.
> > >
> > > **R-Q1.** Thank you for pressing on this point. We argue that the LLM-friendly tactic library is not merely a workaround for a non-interactive setup. Empirically, Figure 2 shows that its advantage is not confined to the initial proof: it improves the entire repair curve as the repair budget increases, where interaction happens (just not tactic-by-tactic).
> > >
> > > The theoretical reason is that the granular tactic design provides more opportunities for correction. Consider the goal `sll(NULL, l) |-- sll(NULL, l) && [|length(l) = 0|]`: cancelling `sll(NULL, l)` without first deriving `l = nil` via `prop_apply` makes the pure subgoal unprovable. LLM-friendly tactics offer more correction opportunities through atomic steps: `split_pure_spatial` and `dump_pre_spatial` isolate and focus on the pure proof step by step, while `entailer!` collapses this into one step. The more atomic design gives the model more intermediate states to detect a wrong path.
> > >
> > > Also, these two tactic suits are not mutually exclusive: one can still use the original tactics for aggressive automation to quickly close goals.
> > >
> > > Finally, regarding generalizability, QCP and VST are very similar in their framework design, and our method can apply to VST and Iris. In particular, the `entailer!` problem we identify is equally present in VST, since VST also relies on `entailer!` for goal simplification.

---

### Official Review · Reviewer_n52k · 2026-03-13

**Soundness:** 3
**Presentation:** 3
**Significance:** 3
**Originality:** 3
**Overall Recommendation:** 4
**Confidence:** 3

**Summary:**

This paper addresses an overlooked challenge in formal verification of system software: standard SMT solvers often fail to automatically prove separation logic verification conditions (VCs) involving complex heap manipulations. Even when correct loop invariants are provided, engineers still need to construct proofs largely by hand. The authors make three contributions. First, they construct the SL-VC benchmark, a suite of 208 separation logic verification conditions extracted from real-world C programs — including the doubly linked list library from the LiteOS kernel and the mini-gmp library — specifically designed to evaluate the deductive reasoning capabilities of AI models. Second, they design an LLM-friendly tactic library for Rocq, an interactive theorem prover. Unlike conventional tactics that encapsulate large amounts of reasoning inside a black box, this library decomposes reasoning into explicit, fine-grained operational steps, enabling LLMs to reason predictably through chain-of-thought. Third, they propose the SPLIT framework (Split spatial and pure Proving with LLM-frIendly Tactics), which employs a two-stage workflow that structurally separates spatial reasoning from pure condition solving, aligning with the semantic structure of separation logic itself.

**Compliance With Llm Reviewing Policy:**

Affirmed.

**Final Justification:**

The authors addressed my concerns as follows:

1. Benchmark scale: They clarify this is an initial benchmark for a new direction (like early miniF2F). The smaller size is justified by the difficulty of collecting separation logic data with manual proofs and annotations, and it already shows clear model differentiation.

2. Performance gap: Analysis of conversation logs shows DeepSeek's advantage comes mainly from the spatial stage (92.8% vs ~65% for others), not the pure stage. DeepSeek settles on viable transformation sequences early and makes local repairs, while Qwen/Kimi often change direction or get stuck on matching errors.

3. Tactic methodology: The library is designed around predictable proof-state changes, derived from expert proof traces (spatial-first strategies) and functional decomposition of existing tactics, validated through ablation studies.

Overall, my concerns have been addressed. I support acceptance and look forward to future extensions of this benchmark.

**Key Questions For Authors:**

1. Table 3 reveals a substantial performance gap between DeepSeek-V3.2 (73.4%) and both Qwen3-235B-A22B and Kimi-K2-Thinking (~51%). Could the authors provide deeper analysis regarding the source of this disparity—for instance, whether it stems from differences in model architecture, context window utilization, or specific capabilities in handling spatial reasoning versus pure logical deduction?

2. What methodology was employed to identify and validate the LLM-friendly tactics? It would be helpful to understand whether these tactics were derived through empirical analysis of model failure modes, manual inspection of expert proof traces, or an iterative refinement process, and whether any tactics were excluded during the design phase.

**Limitations:**

yes

**Strengths And Weaknesses:**

## Strengths

1. The proposed LLM-friendly tactic library and the SPLIT two-stage workflow demonstrate measurable improvements over baseline methods, with the ablation studies providing clear evidence of each component's contribution.

2. The benchmark construction is rigorous: all functions are formally verified by experts who provide precise annotations and manually discharge the resulting verification conditions.

3. The LLM-friendly tactic design addresses a concrete technical gap—unlike human experts who can handle opaque automation, LLMs require explicit, granular proof steps with predictable state transitions, and the tactic library effectively accommodates this constraint.

## Weaknesses:

1. While the 208 verification conditions are derived from real-world system code, the benchmark scale remains relatively modest compared to broader program verification datasets, potentially limiting the diversity of heap manipulation patterns evaluated.

2. The dataset focuses primarily on linked list operations and foundational data structures (e.g., binary search trees, GMP library functions). The framework's generalization to more complex heap structures, such as arbitrary graph topologies, concurrent data structures, or nested recursive structures, has yet to be demonstrated.

---

> ### Author Rebuttal · Authors · 2026-03-30
>
> We thank Reviewer n52k for the positive assessment of our tactic library design and benchmark construction. We address the weaknesses and questions below.
>
> > Weakness 1 & 2. Benchmark scale is modest; dataset focuses on linked lists and foundational data structures, generalization to more complex heap structures not demonstrated.
>
> R1. We acknowledge that the current benchmark is modest in scale. However, this is expected for a first benchmark in a new direction. Some benchmarks in new areas start small (e.g., miniF2F with 488 problems in its initial release, LIG-MM [1] with 312 programs for loop invariant generation with memory manipulation). Compared to larger VC benchmarks such as NTP4VC, whose VCs do not involve separation logic or heap reasoning, our data is closer to real programs and substantially harder to collect: each instance requires manual annotations (spatial and pure specifications, loop invariants, auxiliary assertions), the relevant libraries, and a manually written and checked Rocq proof. Despite its size, SL-VC already reveals clear performance differentiation across models and VC categories, demonstrating its value for evaluating LLM-based separation-logic reasoning.
>
> > Question 1. Table 3 reveals a substantial performance gap between DeepSeek-V3.2 (73.4%) and both Qwen3-235B-A22B and Kimi-K2-Thinking (~51%). Could the authors provide deeper analysis?
>
> R2. We examined conversation logs on representative cases. DeepSeek V3's advantage mainly comes from the spatial stage, not from a general advantage on the pure stage:
>
> | Model | Spatial pass rate | Pure pass rate* |
> | --- | --- | --- |
> | DeepSeek V3 | 192/207 (92.8%) | 81/121 (66.9%) |
> | Qwen | 140/207 (67.6%) | 47/81 (58.0%) |
> | Kimi | 134/207 (64.7%) | 55/81 (67.9%) |
>
> *Pure pass rates are computed only over goals that actually reached the pure stage with nontrivial pure subtasks. Denominators differ because goals where the spatial stage failed or already resolved all obligations do not enter the pure stage.
>
> Reading representative logs (e.g., `bst_delete_rec/get_pre_return_wit_1`, `gmp/mpz_clear_return_wit_1_1`), a consistent pattern is that DeepSeek settles on a workable spatial transformation sequence early and makes only local repairs after verifier feedback, while Qwen and Kimi more often keep changing tactic direction or get stuck on local matching errors.
>
> > Question 2. What methodology was employed to identify and validate the LLM-friendly tactics?
>
> R3. Our design follows one central principle: proof-state changes caused by a tactic should be predictable. We derived the library from two sources: (1) analysis of expert proof traces, where humans typically first perform spatial transformations via `sep_apply`, then use `entailer!` to eliminate most spatial predicates, and finally discharge remaining goals; and (2) functional analysis of existing tactics: decomposing multi-effect tactics like `entailer!` and refining tactics like `sep_apply` that introduce pure side effects. The ablation study validates this design.
>
> [1] Towards General Loop Invariant Generation: A Benchmark of Programs with Memory Manipulation.

---

> > ### Author Rebuttal · Reviewer_n52k · 2026-04-04
> >
> > Thank you for the response. My questions have been addressed, and I am inclined to raise my score to 4.

---

### Decision · Program_Chairs · 2026-04-30

**Decision:**

Accept (regular)

**Comment:**

Reviewers agreed that this paper tackles an interesting and often overlooked problem in LLM-supported formal verification of software systems. The paper contributes a well crafted benchmark showing the limitations of current approaches, and proposes a new tactic library that can be used to alleviate these limitations, as shown in several use cases. The reviewers and the area chair agree that the results of this paper are solid contribution for the ICML community, and recommend acceptance.